# Serum proteomic profiling of physical activity reveals CD300LG as a novel exerkine with a potential causal link to glucose homeostasis

Sindre Lee-Ødegård[1,2]*, Marit Hjorth[3†], Thomas Olsen[3†], Gunn-Helen Moen[2,4,5,6], Emily Daubney[4], David M Evans[4,6,7], Andrea L Hevener[8], Aldons J Lusis[9,10], Mingqi Zhou[11], Marcus M Seldin[11], Hooman Allayee[12,13], James Hilser[12,13], Jonas Krag Viken[2], Hanne Gulseth[14], Frode Norheim[3], Christian A Drevon[15], Kåre Inge Birkeland[1,2]

[1]Department of Endocrinology, Morbid Obesity and Preventive Medicine, Oslo University Hospital, Oslo, Norway; [2]Institute of Clinical Medicine, Faculty of Medicine, University of Oslo, Oslo, Norway; [3]Department of Nutrition, Institute of Basic Medical Sciences, Faculty of Medicine, University of Oslo, Oslo, Norway; [4]Institute for Molecular Bioscience, The University of Queensland, Brisbane, Australia; [5]The Frazer Institute, The University of Queensland, Woolloongabba, Australia; [6]Department of Public Health and Nursing, K.G. Jebsen Center for Genetic Epidemiology, NTNU, Norwegian University of Science and Technology, Trondheim, Norway; [7]MRC Integrative Epidemiology Unit, University of Bristol, Bristol, United Kingdom; [8]Division of Endocrinology, Department of Medicine, David Geffen School of Medicine, University of California, Los Angeles, Los Angeles, United States; [9]Department of Human Genetics, University of California, Los Angeles, Los Angeles, United States; [10]Division of Cardiology, Department of Medicine, David Geffen School of Medicine at UCLA, Los Angeles, United States; [11]Department of Biological Chemistry, University of California, Irvine, Irvine, United States; [12]Departments of Population and Public Health Sciences, Keck School of Medicine, University of Southern California, Los Angeles, United States; [13]Department of Biochemistry and Molecular Medicine, Keck School of Medicine, University of Southern California, Los Angeles, United States; [14]Department of Chronic Diseases and Ageing, Norwegian Institute of Public Health, Oslo, Norway; [15]Vitas Ltd, Oslo, Norway

*For correspondence:
sindre.lee@medisin.uio.no

†Second authorship

## Abstract

**Background:** Physical activity has been associated with preventing the development of type 2 diabetes and atherosclerotic cardiovascular disease. However, our understanding of the precise molecular mechanisms underlying these effects remains incomplete and good biomarkers to objectively assess physical activity are lacking.

**Methods:** We analyzed 3072 serum proteins in 26 men, normal weight or overweight, undergoing 12 weeks of a combined strength and endurance exercise intervention. We estimated insulin sensitivity with hyperinsulinemic euglycemic clamp, maximum oxygen uptake, muscle strength, and used MRI/MRS to evaluate body composition and organ fat depots. Muscle and subcutaneous adipose tissue biopsies were used for mRNA sequencing. Additional association analyses were performed

in samples from up to 47,747 individuals in the UK Biobank, as well as using two-sample Mendelian randomization and mice models.

**Results:** Following 12 weeks of exercise intervention, we observed significant changes in 283 serum proteins. Notably, 66 of these proteins were elevated in overweight men and positively associated with liver fat before the exercise regimen, but were normalized after exercise. Furthermore, for 19.7 and 12.1% of the exercise-responsive proteins, corresponding changes in mRNA expression levels in muscle and fat, respectively, were shown. The protein CD300LG displayed consistent alterations in blood, muscle, and fat. Serum CD300LG exhibited positive associations with insulin sensitivity, and to angiogenesis-related gene expression in both muscle and fat. Furthermore, serum CD300LG was positively associated with physical activity and negatively associated with glucose levels in the UK Biobank. In this sample, the association between serum CD300LG and physical activity was significantly stronger in men than in women. Mendelian randomization analysis suggested potential causal relationships between levels of serum CD300LG and fasting glucose, 2 hr glucose after an oral glucose tolerance test, and HbA1c. Additionally, Cd300lg responded to exercise in a mouse model, and we observed signs of impaired glucose tolerance in male, but not female, *Cd300lg* knockout mice.

**Conclusions:** Our study identified several novel proteins in serum whose levels change in response to prolonged exercise and were significantly associated with body composition, liver fat, and glucose homeostasis. Serum CD300LG increased with physical activity and is a potential causal link to improved glucose levels. CD300LG may be a promising exercise biomarker and a therapeutic target in type 2 diabetes.

**Funding:** South-Eastern Norway Regional Health Authority, Simon Fougners Fund, Diabetesforbundet, Johan Selmer Kvanes' legat til forskning og bekjempelse av sukkersyke. The UK Biobank resource reference 53641. Australian National Health and Medical Research Council Investigator Grant (APP2017942). Australian Research Council Discovery Early Career Award (DE220101226). Research Council of Norway (Project grant: 325640 and Mobility grant: 287198). The Medical Student Research Program at the University of Oslo. Novo Nordisk Fonden Excellence Emerging Grant in Endocrinology and Metabolism 2023 (NNF23OC0082123).

**Clinical trial number:** clinicaltrials.gov: NCT01803568.

## eLife assessment

This **useful** article describes a proteomic analysis of plasma from subjects before and after an exercise regime consisting of endurance and resistance exercise. The work identifies a putative new exerkine, CD300LG, and finds associations of this protein with aspects of insulin sensitivity and angiogenesis. The characterization remains **incomplete** at present. Because CD300LG may have a transmembrane domain, one possibility is that exercise causes the release of extracellular vesicles containing this protein. As this study reports associations, additional studies will be needed to establish causality. The article will hopefully prompt further studies to more fully elucidate the underlying biology.

## Introduction

Physical activity is a cornerstone in the prevention and treatment of several chronic diseases like obesity, non-alcoholic fatty liver disease (NAFLD), atherosclerotic vascular disease, and type 2 diabetes mellitus (*Piercy et al., 2018*). Both acute- and long-term exercise may enhance insulin sensitivity and thereby improve glucose tolerance (*Hawley and Lessard, 2008*). Both resistance and endurance exercises enhance insulin sensitivity, although the most pronounced effect is observed when combining these training modalities (*Bacchi et al., 2012*).

Metabolic adaptations to exercise encompass intricate inter-organ communication facilitated by molecules referred to as exerkines (*Pedersen et al., 2007*; *Chow et al., 2022*; *Görgens et al., 2015*; *Lee-Ødegård et al., 2022*). These exerkines are secreted from various tissues and include a variety of signal molecules released in response to acute- and/or long-term exercise with endocrine, paracrine, and/or autocrine functions (*Pedersen et al., 2007*; *Chow et al., 2022*). Although there has been considerable emphasis on exerkines originating from skeletal muscle (SkM) (*Pourteymour et al.,*

*2017*; *Pedersen and Febbraio, 2012*), it is also known that exerkines can originate from organs such as white (*Görgens et al., 2015*; *Lee et al., 2019*; *Bouassida et al., 2010*) and brown adipose tissue (*Stanford et al., 2018*) or the liver (*Lee et al., 2017*). The established bona fide exerkine, interleukin-6 (IL6), is released during muscle contractions, contributing to improved overall glucose homeostasis (*Pedersen et al., 2007*; *Kistner et al., 2022*). In addition, a range of other exerkines are recognized, including IL7 (*Haugen et al., 2010*), 12,13-diHOME (*Stanford et al., 2018*), myonectin (*Otaka et al., 2018*), myostatin (*Hjorth et al., 2016*; *McPherron et al., 1997*), METRNL (*Rao et al., 2014*), CSF1 (*Pourteymour et al., 2017*), decorin (*Kanzleiter et al., 2014*), SFRP4 *Lee et al., 2019*, fetuin-A (*Lee et al., 2017*; *Malin et al., 2014*), and ANGPTL4 (*Catoire et al., 2014*; *Norheim et al., 2014*), among many others (*Chow et al., 2022*).

Extensive screening aimed at discovering novel exercise responsive blood proteins have faced considerable challenges, primarily due to the technical challenges in quantifying the blood proteome on a large scale. However, recent advances in multi-plex technology, such as the proximity extension assay (PEA), have made it possible to quantify more than 3000 proteins in blood samples more reliably than traditional untargeted mass spectrometry (https://olink.com/technology/what-is-pea). Some recent studies have used other proteomic platforms, such as aptamer-based techniques (https://somalogic.com/somascan-platform/), to show that acute- and long-term aerobic exercise affected several hundred serum proteins (*Contrepois et al., 2020*; *Diaz-Canestro et al., 2023*; *Robbins et al., 2021*; *Robbins et al., 2023*; *Mi et al., 2023*), but the downstream causal effects of such changes on clinical phenotypes are not known. Furthermore, no studies have used the PEA technology to identify exerkines potentially underlying the mechanisms through which long-term physical activity, including strength exercise, enhances glucose homeostasis.

We performed the 'physical activity, myokines, and glucose metabolism' (MyoGlu) study (*Langleite et al., 2016*), which was a controlled clinical trial aiming to identify novel secreted factors ('exerkines') that may serve as links between physical activity and glucose metabolism. We conducted a comprehensive serum screening of 3072 proteins in normal weight and overweight men both before and after combined endurance and strength exercise. Rigorous phenotyping was carried out, including hyperinsulinemic euglycemic clamping, assessments of maximum oxygen uptake, maximum muscle strength, and ankle-to-neck MRI/MRS scans.

Exerkines identified with potential effects on glucose homeostasis in the MyoGlu study were subsequently subject to analysis across several external data sets. Using data from 47,747 participants in the UK Biobank (*Sudlow et al., 2015*), we assessed correlations between candidate proteins and estimates of physical activity and glucometabolic outcomes. These associations were then tested for causality using Mendelian randomization (MR). Exerkines of interest were also assessed in a knockout mouse model and in a exercise mouse model to further assess potential links with glucose homeostasis.

## Methods

The MyoGlu study was conducted as a controlled clinical trial (clinicaltrials.gov: NCT01803568) and was carried out in adherence to the principles of the Declaration of Helsinki. The study received ethical approval from the National Regional Committee for Medical and Health Research Ethics North in Tromsø, Norway, with the reference number 2011/882. All participants provided written informed consent before undergoing any procedures related to the study. The UK Biobank has ethical approval from the North West Multi-Centre Research Ethics Committee (MREC), which covers the UK, and all participants provided written informed consent. This particular project from the UK Biobank received ethical approval from the Institutional Human Research Ethics committee, University of Queensland (approval number 2019002705).

### Participants

The MyoGlu study enrolled men aged 40–65 years who were healthy but sedentary (having engaged in fewer than one exercise session per week in the previous year) (*Langleite et al., 2016*; *Lee et al., 2021*). These participants were divided into two groups based on their body mass index (BMI) and glucose tolerance: overweight (with an average BMI of 29.5 ± 2.3 kg/m$^2$) and normal weight controls (with an average BMI of 23.6 ± 2.0 kg/m$^2$). The overweight men had reduced glucose tolerance and/or

insulin sensitivity (*Supplementary file 1A*). Both groups, consisting of 13 individuals each, underwent a 12-week regimen of combined strength and endurance training.

## Exercise protocols

This 12-week training intervention included two weekly sessions of 60 min each for endurance cycling and two sessions of 60 min each for whole-body strength training. Blood samples, and muscle (*m. vastus lateralis*) and subcutaneous white adipose tissue biopsies were taken at baseline before the intervention, and then again at least 3 days after the last exercise session of the 12-week intervention period (*Langleite et al., 2016*; *Lee et al., 2021*).

## Clinical data

The euglycemic hyperinsulinemic clamp was performed after an overnight fast (*Langleite et al., 2016*; *Lee et al., 2021*). A fixed dose of insulin 40 mU/m$^2$·min$^{-1}$ was infused, and glucose (200 mg/mL) was infused to maintain euglycemia (5.0 mmol/L) for 150 min. Insulin sensitivity is reported as glucose infusion rate (GIR) during the last 30 min relative to body weight. Whole blood glucose concentration was measured using a glucose oxidase method (YSI 2300, Yellow Springs, OH) and plasma glucose concentration was calculated as whole blood glucose × 1.119. Magnetic resonance imaging/spectroscopy (MRI/MRS) methods were used to quantify fat and lean mass. The ankle-to-neck MRI protocol included a 3D DIXON acquisition providing water and lipid quantification, data were then analyzed using the nordicICE software package (NordicNeuroLab, Bergen, Norway), and the jMRUI workflow. VO$_2$max tests were performed after standardized warm-up at a workload similar to the final load of an incremental test in which the relationship between workload (Watt) and oxygen uptake was established. Participants cycled for 1 min followed by a 15-Watt increased workload every 30 s until exhaustion. Test success was based on O$_2$ consumption increased <0.5 mL·kg$^{-1}$·min$^{-1}$ over a 30-Watt increase in workload, respiratory exchange ratio values >1.10, and blood lactate >7.0 mmol/L. We obtained scWAT, SkM biopsies, and blood samples as described previously (*Langleite et al., 2016*). Biopsies were obtained from the periumbilical subcutaneous tissue and from *m. vastus lateralis*. After sterilization, a lidocaine-based local anesthetic was injected in the skin and sub cutis prior to both SkM and scWAT biopsies. Biopsies were dissected on a cold aluminium plate to remove blood, etc., before freezing. For standard serum parameters, measurement were either conducted using standard in-house methods or outsourced to a commercial laboratory (Fürst Laboratories, Oslo, Norway).

## The Olink proteomics explorer 3072 platform

We utilized antibody-based technology (Olink Proteomics AB, Uppsala, Sweden) to conduct profiling of serum samples through the Olink Explore 3072 panel. This PEA technique involves using pairs of DNA oligonucleotide-labeled antibodies to bind to the proteins of interest. When two matching antibodies attach to a target protein, the linked oligonucleotides hybridize and are extended by DNA polymerase, forming a unique DNA 'barcode'. This barcode is then read using next-generation sequencing. The specificity and sensitivity of the PEA technology are notably high because only accurately matched DNA pairs generated detectable and measurable signals. To refine the dataset, proteins that were not detected or were duplicated were removed, resulting in an analysis of 2886 proteins. Only a single assay was conducted, eliminating inter-assay variability. Data are presented as normalized protein expression (NPX) units, which are logarithmically scaled using a log$_2$ transformation.

## Proteomics validations

Duplicate measurements of IL6 and leptin in plasma were conducted using ELISA kits (Leptin; Camarillo, CA; and IL6; R&D Systems, Minneapolis, MN) following the manufacturer's instructions. The correlations between PEA or ELISA assays were *r* = 0.94 (p=1.4 × 10$^{-11}$), and *r* = 0.92 (p=2.2 × 10$^{-11}$) for IL6 and leptin, respectively (*Figure 2—figure supplement 1*).

## mRNA sequencing

Biopsies were frozen in liquid nitrogen, crushed to powder by a pestle in a liquid nitrogen-cooled mortar, transferred into 1 mL QIAzol Lysis Reagent (QIAGEN, Hilden, Germany), and homogenized using TissueRuptor (QIAGEN) at full speed for 15 s, twice (*Langleite et al., 2016*; *Lee et al., 2021*). Total RNA was isolated from the homogenate using miRNeasy Mini Kit (QIAGEN). RNA integrity and

concentration were determined using Agilent RNA 6000 Nano Chips on a Bioanalyzer 2100 (Agilent Technologies Inc, Santa Clara, CA). RNA was converted to cDNA using High-Capacity cDNA Reverse Transcription Kit (Applied Biosystems, Foster, CA). The cDNA reaction mixture was diluted in water and cDNA equivalent of 25 ng RNA used for each sample. All muscle and scWAT samples were deep-sequenced using the Illumina HiSeq 2000 system with multiplex at the Norwegian Sequencing Centre, University of Oslo. Illumina HiSeq RTA (real-time analysis) v1.17.21.3 was used. Reads passing Illumina's recommended parameters were demultiplexed using CASAVA v1.8.2. For prealignment quality checks, we used the software FastQC v0.10.1. The mean library size was ~44 millions unstranded 51 bp single-ended reads for muscle tissue and ~52 millions for scWAT with no differences between groups or time points. No batch effects were present. cDNA sequenced reads alignment was done using Tophat v2.0.8, Samtools v0.1.18, and Bowtie v2.1.0 with default settings against the UCSC hg19 annotated transcriptome and genome dated May 14, 2013. Post-alignment quality controls were performed using the Integrative Genome Viewer v2.3 and BED tools v2.19.1. Reads were counted using the intersection strict mode in HTSeq v0.6.1.

## Statistics and bioinformatics

Olink data were analyzed using the 'AnalyzeOlink' R package for pre-processing, testing using mixed linear regression and annotation. Pathway and Gene Ontology overrepresentation analyses were performed using MSigDB data sets (Hallmark pathways and biological processes). mRNA sequencing data were normalized as reads per kilobase per million mapped read (RPKM) and analyzed using mixed linear regression from the 'lme4' R package. Normality was determined by quantile–quantile plots. p-values were corrected using the Benjamini–Hochberg approach set at a false discovery rate (FDR) of 5%. For univariate correlations, Pearson's or Spearman's method was applied as appropriate. Principal component analysis was performed using the 'prcomp' R package. Key driver analysis was performed using the 'Mergeomics' R package. Mediation analysis was performed using the 'Mediation' R package with 1000 bootstraps and the *set.seed* function to ensure reproducibility.

## UK Biobank

The UK Biobank is a large prospective population-based cohort containing ~500,000 individuals (~273,000 women), with a variety of phenotypic and genome-wide genetic data available (*Sudlow et al., 2015*). The UK Biobank has ethical approval from the North West Multi-Centre Research Ethics Committee (MREC), which covers the UK, and all participants provided written informed consent.

We utilized imputed genetic data from the October 2019 (version 3) release of the UK Biobank for our analyses (application ID: 53641). In addition to the quality control metrics performed centrally by the UK Biobank (*Bycroft et al., 2018*), we defined a subset of unrelated 'white European' individuals. We excluded those with putative sex chromosome aneuploidy, high heterozygosity or missing rate, or a mismatch between submitted and inferred sex as identified by the UK Biobank (total N = 1932). We excluded individuals who we did not identify as ancestrally European using K-means clustering applied to the first four genetic principal components generated from the 1000 Genomes Project (*1000 Genomes Project Consortium et al., 2015*). We also excluded individuals who had withdrawn their consent to participate in the study as of February 2021.

## The Olink proteomics explorer 1536 platform

All analysis were done using the UK Biobank Olink data containing a total of 58,699 samples and 54309 individuals, after excluding individuals as mentioned above we had 47,747 samples with measured serum CD300LG levels. Data was generated according to Olink's standard procedures.

## Observational analyses

For the physical activity measurements, we investigated if the degree of physical activity was associated with serum levels of protein (serum levels of protein regressed on physical activity); alternatively for the metabolic measurements we investigated if the protein expression affected the metabolic measurements (trait regressed on serum levels of protein), for both we used a linear regression model. We performed analyses stratified by sex and adjusting for age, protein batch, UK Biobank assessment centre, the time the sample was stored and BMI. All analyses were performed in R version 3.4.3.

## Genome-wide association analysis

A GWAS of serum CD300LG levels ($log_2$ transformed) measured in the UK Biobank was performed using BOLT-LMM (*Loh et al., 2015*) on individuals of European descent who had proteomic data available (N = 45,788). We included sex, year of birth, protein and genotyping batch, time from sample collection to processing time (in weeks), and five ancestry informative principal components as covariates in the analysis.

Post GWAS quality control included the removal of SNPs with (minor allele frequency) MAF ≤ 0.05 and info score ≤0.4 ($n_{SNPs}$ = 6,945,819). The previously generated LD reference panel for clumping consisted of a random sample of 47,674 unrelated British UK Biobank individuals identified using GCTA (*Wu et al., 2022*) with identity by state (IBS) < 0.025 and identity by descent (IBD) sharing of <0.1. LD score regression analysis (*Lee et al., 2018*) was used to investigate whether genomic inflation was likely due to polygenicity or population stratification/cryptic relatedness.

Prior to gene annotation, palindromic SNPs were excluded ($n_{SNPs}$ = 6,882,889 remaining). Variants were classified as either *cis*- or *trans*-pQTLs based on SNP proximity to the protein-encoding gene (CD300LG) of interest. Variant annotation was performed using ANNOVAR (*Wang et al., 2010*), labeling genes ±500 kb from variants. A pQTL was considered a *cis*-pQTL if the gene annotation in the 1 Mb window matched the protein name, all remaining variants were considered *trans*-pQTLs.

To extract independent genome-wide significant pQTLs ($p<5 \times 10^{-8}$), clumping was performed using the PLINK v1.90b3.31 software package (*Purcell et al., 2007*); variants with $r^2 > 0.001$ with the index SNP were removed using a 1 Mb window. Variants that lied within the human major histocompatibility complex region were removed, excluding pQTLs on chromosome 6 from 26 Mb to 34 Mb.

## Mendelian randomization

To obtain valid instrumental variables (SNPs) for our analysis, we assessed them against the three core assumptions for MR analysis: (1) that the SNPs were robustly associated with the exposure of interest. For that, we obtained summary result statistics on genome-wide significant SNPs from our own GWAS. (2) That the SNPs were not associated with any known or unknown confounders. This is not an assumption that can be fully tested; however, we used PhenoScanner (*Staley et al., 2016*; *Kamat et al., 2019*) to assess whether any SNPs were associated with known confounders (described below). (3) That the SNPs were not associated with the outcomes through any other path than through the exposure. To test this assumption, we searched PhenoScanner (*Staley et al., 2016*; *Kamat et al., 2019*) (detailed below) to see if our exposures of interest were associated with other potentially pleiotropic phenotypes.

## MR statistical analysis

We used the TwoSampleMR package (*Hemani et al., 2018*; https://github.com/MRCIEU/TwoSampleMR; *Palmer and Hemani, 2024*) in R version 4.2.2 (https://cran.r-project.org/). The outcome studies were obtained from http://magicinvestigators.org/ (*Chen et al., 2021*) and were external to the UK Biobank. Specifically, we used the outcomes 'fasting glucose adjusted for BMI' (mmol/L, n = 200,622), '2 hr post OGTT glucose adjusted for BMI' (mmol/L, n = 63,396), 'fasting insulin adjusted for BMI' (pmol/L, n = 151,013), and 'HbA1c' (%, n = 146,806) (*Chen et al., 2021*).

We performed a two-sample inverse variance weighted (IVW) analysis to assess the causal effect of CD300LG on metabolic factors (*Supplementary file 1H and I*). To explore potential violations of the core assumptions when using the full set of SNPs, we performed a heterogeneity test using Cochran's Q, and a test for directional pleiotropy was conducted by assessing the degree to which the MR Egger intercept differed from zero (*Bowden et al., 2015*). We also performed additional sensitivity analyses using MR Egger regression (*Bowden et al., 2015*), weighted median (*Bowden et al., 2016*), simple and weighted mode estimation methods (*Hartwig et al., 2017*). Effect estimates from the different sensitivity analysis were compared as a way of assessing the robustness of the results. To assess potential heterogeneity in the MR estimates, we further performed MR-PRESSO (*Chen et al., 2021*; *Verbanck et al., 2018*) to detect (MR-PRESSO global test) and correct for horizontal pleiotropy via outlier removal (MR-PRESSO outlier test).

## Investigation of potentially pleiotropic SNPs

SNPs robustly associated with the exposure investigated in the MR analyses (serum CD300LG levels) were checked for other possible associations (PhenoScanner v2; *Staley et al., 2016*; *Kamat et al., 2019*, http://www.phenoscanner.medschl.cam.ac.uk/), which may contribute to a pleiotropic effect on the metabolic outcomes. *Supplementary file 1J* lists the SNPs used in our analysis that many influence related phenotypes. Phenotypes from PhenoScanner were listed if they were associated with the SNPs or nearby variants in high LD ($r^2 = 0.8$) at p-value level $<1 \times 10^{-5}$ and could have potential pleiotropic effects in the analysis.

## Results

### Cohort characteristics

We studied 26 male participants, including 13 with normal weight, and another 13 with overweight, as described previously (*Langleite et al., 2016*). They were subjected to 12 weeks of high-intensity resistance and endurance exercise (*Figure 1*). The overweight participants had lower glucose tolerance and insulin sensitivity compared to the normal weight participants (*Supplementary file 1A*). After the 12-week intervention, body fat mass decreased and lean body mass increased, together with significant improvements in insulin sensitivity (~40%), maximum oxygen uptake and muscle strength in both groups (*Supplementary file 1A*).

### Serum proteome responses to prolonged exercise

Recognizing that circulating proteins could mediate exercise-induced metabolic improvements, we next investigated alterations in the serum proteome in response to the 12-week intervention using PEA technology. Of the 3072 proteins quantified, we detected increased serum concentrations of 126 proteins and decreased serum concentrations of 157 proteins following the 12-week intervention at an FDR below 5% (*Figure 2A–C*; *Supplementary file 1B–D*). Among these, 20 proteins increased exclusively in normal weight men, whereas 19 proteins increased exclusively in overweight men (*Figure 2D*). Four proteins were uniquely reduced in normal weight men, and 66 proteins were uniquely reduced in overweight men (*Figure 2E*).

Several of the exercise-responsive proteins had potential roles in muscle adaptation and metabolism. For example, platelet-derived growth factor subunit B (PDGFB) and IL7 are both myokines with potential effects on muscle differentiation (*Haugen et al., 2010*; *Hamaguchi et al., 2023*). Further, fibroblast growth factor-binding protein 3 (FGFBP3) may influence running capacity (*Lories et al., 2009*) and muscle strength (*Casas-Fraile et al., 2020*). NADH-cytochrome b5 reductase 2 (CYB5R2) can preserve SkM mitochondria function in aging mice (*López-Bellón et al., 2022*). FGFBP3 and switch-associated protein 70 (SWAP70) may protect against weight gain (*Tassi et al., 2018*) and cardiac hypertrophy (*Qian et al., 2023*), respectively. Finally, dual specificity protein phosphatase 13 isoform A (DUSP13A) is highly specific to SkM (*Chen et al., 2004*), making it a potential novel muscle-specific marker for long-term exercise. Detailed results for 2885 proteins in response to prolonged exercise are shown in *Supplementary file 1B*.

### A proteomic liver fat signature in overweight men

In response to the 12-week exercise intervention, a larger number of serum proteins responded in overweight men than in normal weight men (*Figure 2B and C*). In particular, 66 proteins decreased in serum after 12 weeks in overweight men (*Figure 2E*). Gene Ontology analyses revealed known pathways only for the proteins that decreased in overweight men (*Figure 3A and B*), and one of the most enriched pathways is related to metabolism of organic acids (*Figure 3C*). A key driver analysis of the 66 proteins identified SLC22A1, a hepatocyte transporter related to liver fat content (*Figure 3D*). Furthermore, the 66 proteins also displayed a 24% overlap with a known human serum proteomic signature of NAFLD (*Figure 3E*; *Govaere et al., 2023*), but no common proteins with signatures of specific liver cells (The Human Liver Cell Atlas: *Aizarani et al., 2019*). Baseline serum protein concentrations in the identified signature of 66 proteins were higher among men with overweight compared to those with normal weight, but were reduced or normalized in overweight men following prolonged exercise (*Figure 3F*). Using principal component analysis of the 66 proteins, the first component correlated positively to liver fat content at baseline (*Figure 3G*), but not after prolonged

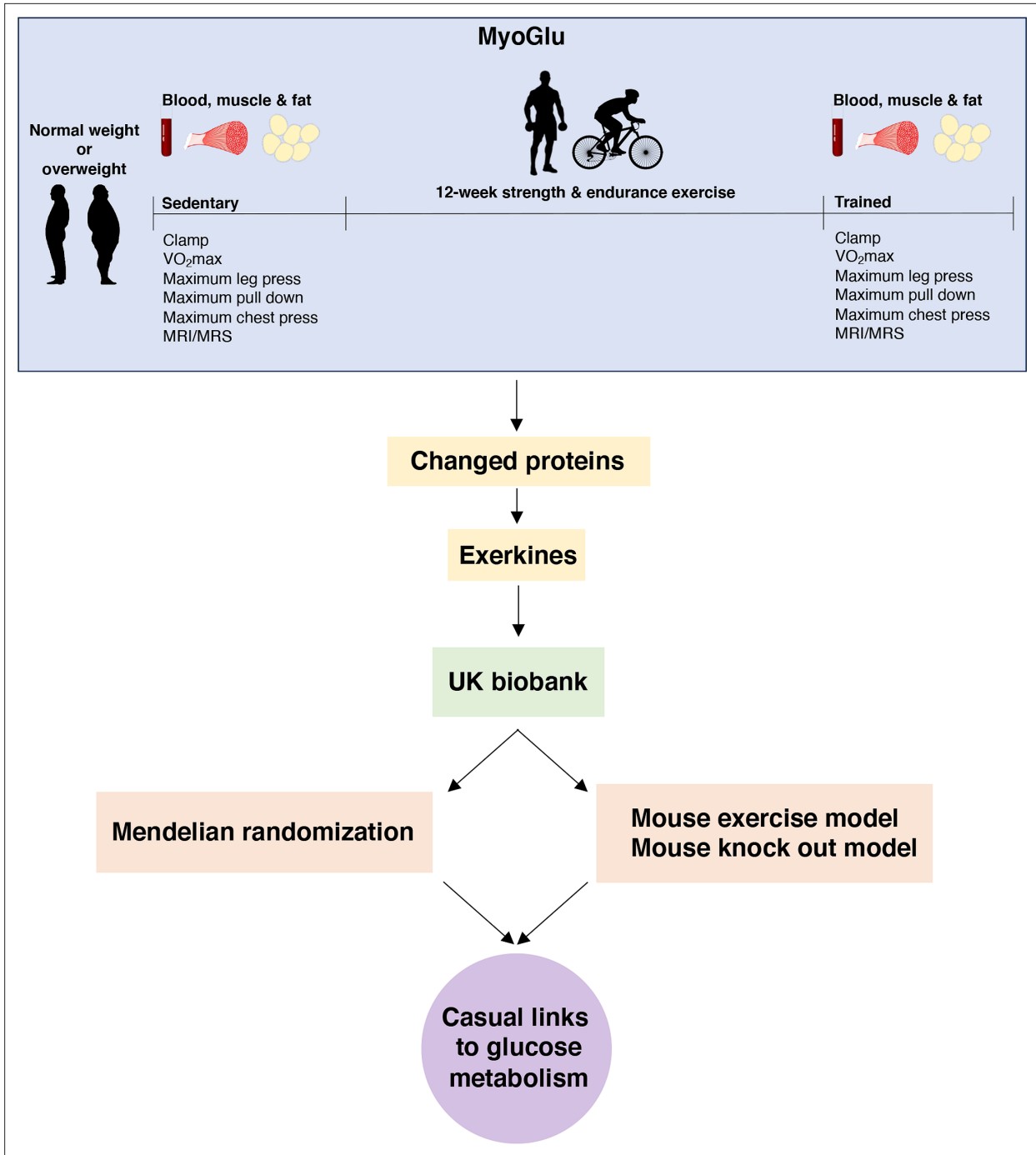

**Figure 1.** Study overview. We recruited sedentary men with either normal weight or overweight for deep phenotyping before and after a prolonged exercise intervention. Multi-omic analyses, including serum proteomics, clinical traits, and muscle and fat transcriptomics, identified changed proteins and potential exerkines. Candidate exerkines were subsequently analyzed in serum samples from the UK Biobank and tested for associations with physical activity and glucometabolic traits. Top candidates were then subjected to Mendelian randomization and investigated in a mouse exercise model and in a mouse knock-out model to assess casual links between exerkines and glucometabolic traits.

exercise (*Figure 3H*). Similarly, the first component also correlated positively with several liver-related markers at baseline (*Figure 3L*) and negatively to insulin sensitivity at baseline (*Figure 3I*), but not after prolonged exercise (*Figure 3J*). The first component mediated 37% of the association between baseline insulin sensitivity and liver fat content (*Figure 3K*). We observed no enrichments for the remaining proteins (*Figure 3A and B*).

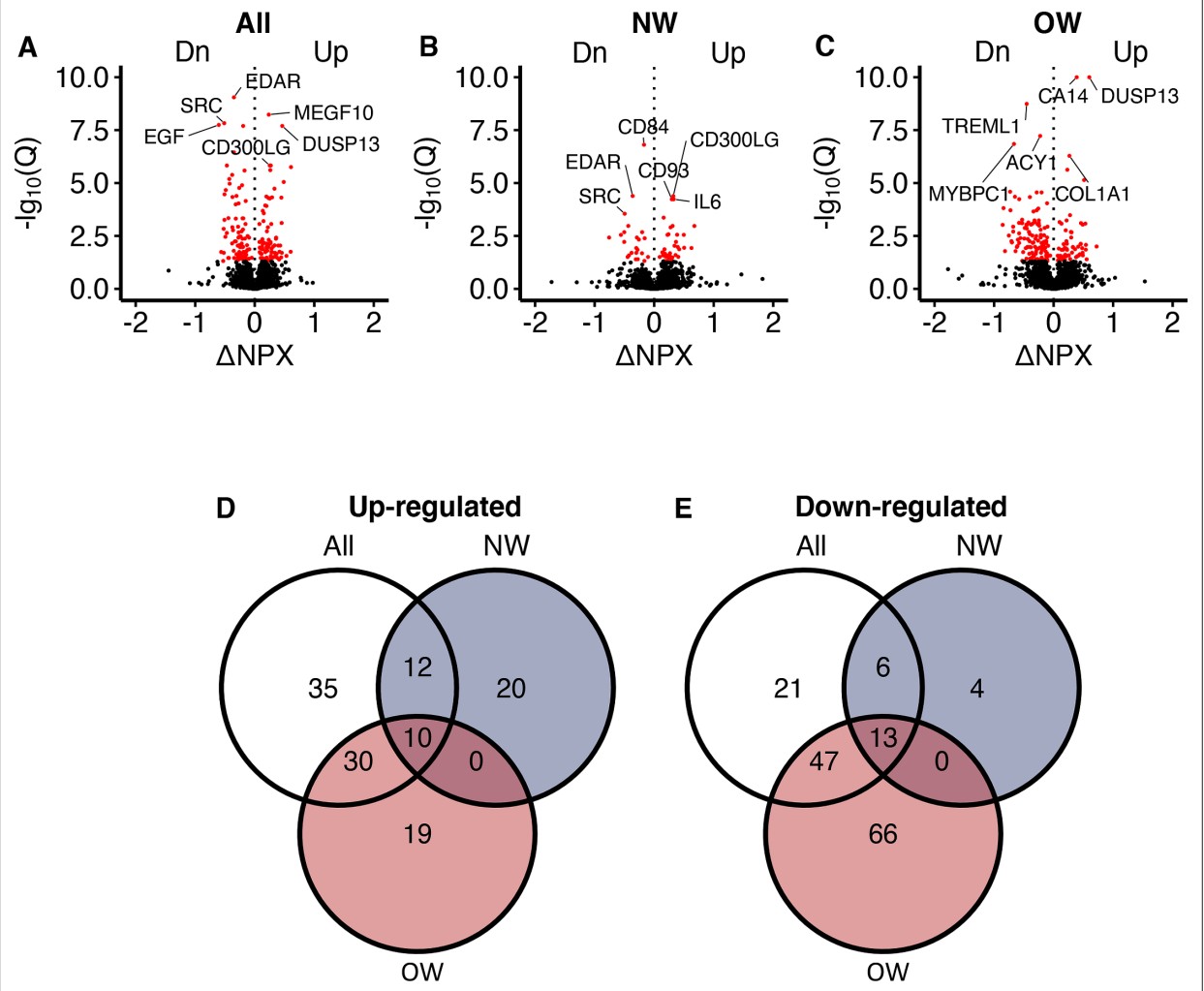

**Figure 2.** Serum proteomic responses to prolonged exercise. (**A**) A volcano plot showing responses in all participants. The x-axis shows log$_2$(fold-changes) and the y-axis shows negative log$_{10}$(Q-values). The red dots indicate statistical significance (Q < 0.05). Only the top three up-/downregulated proteins are annotated. (**B, C**) Similar to (**A**), but in normal weight and overweight men only. (**D, E**) Venn diagrams of the significant change in proteins shown in (**A–C**). NPX = normalized protein expression; Q = p-values corrected using Benjamini–Hochberg's method; NW = normal weight; OW = overweight.

The online version of this article includes the following figure supplement(s) for figure 2:

**Figure supplement 1.** Olink vs. ELISA for (**A**) serum leptin and (**B**) IL6 protein levels.

## Secretory proteins

Among the 96 upregulated and 110 down-egulated serum proteins responding to the 12-week exercise intervention (*Figure 2D and E*), 37 are curated secretory proteins, and another 46 proteins are predicted as highly likely secretory proteins (*Figure 4A–C*). We assessed the corresponding mRNA responses in SkM and subcutaneous white adipose tissue (ScWAT) following the 12-week intervention (*Figure 4A–C*). In total, 19.7% of the serum secretory proteins displayed a directionally consistent significant change mRNA levels in SkM, whereas 12.1% of the serum secretory proteins exhibited a corresponding mRNA response in ScWAT (*Figure 4C*). *COL1A1* was the most responsive SkM mRNA that also had a corresponding increase in serum COL1A1 after prolonged exercise (*Figure 4D and E*). *CCL3* was the most responsive ScWAT mRNA that also had a corresponding decrease in serum after prolonged exercise (*Figure 4F and G*). To prioritize proteins for follow-up analyses, we focused on SMOC1 and CD300LG, which had similar exercise responses in blood, SkM and ScWAT (*Figure 4H*). SMOC1 is a known hepatokine with effects on insulin sensitivity in mice (*Montgomery et al., 2020*),

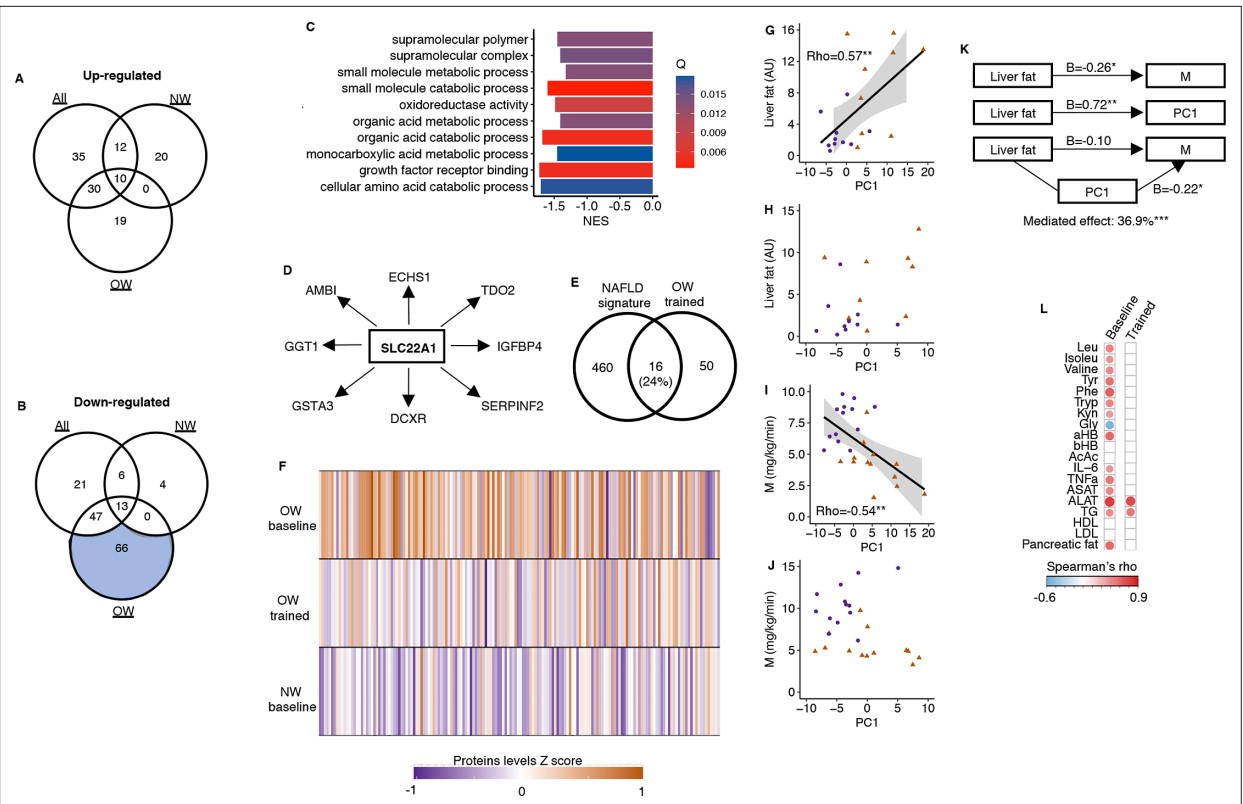

**Figure 3.** A serum proteomic liver fat signature. (**A**) No upregulated proteins after prolonged exercise overlapped with known pathways. (**B**) Only the 66 downregulated proteins in the OW group overlapped with known pathways. (**C**) Top 10 gene sets overlapping with these 66 proteins. (**D**) SLC22A1 is a key driver among these 66 proteins. (**E**) These 66 proteins overlapped with a known human serum proteomic nonalcoholic fatty liver disease signature from *Govaere et al., 2023*. (**F**) The downregulated proteins in the OW group were elevated in OW vs. NW at baseline but normalized in the OW group after prolonged exercise. The principal component of these 66 proteins correlated with (**G**) liver fat content at baseline, but (**H**) not after prolonged exercise, with (**I**) the clamp M value at baseline, but (**J**) not after prolonged exercise. (**K**) The principal component (PC) of these 66 proteins mediated 36.9% of the association between liver fat and M. (**L**) The principal component of these 66 proteins correlated with several liver-related markers at baseline, but not after prolonged exercise except for aspartate transaminase (ASAT) and alanin aminotransferase (ALAT). White = nonsignificant, red/blue = significant. *p<0.05 and **p<0.01.

but probably with no causal link to insulin sensitivity in humans (*Montgomery et al., 2020*; *Ghodsian et al., 2021*). Thus, we focused on CD300LG in subsequent analyses.

## CD300LG

CD300LG displayed increased concentration in serum (+20%, p<0.001) together with increased levels in both SkM (+60%, p<0.001) and scWAT (+13%, p=0.01) mRNA following the 12-week exercise intervention (*Figure 5A–C*). Changes in serum CD300LG correlated positively with changes in insulin sensitivity after the intervention (rho = 0.59, p=0.002; *Figure 5D*). In addition, serum CD300LG concentration was lower in overweight than normal weight men (–51%, p=0.014) and positively correlated with insulin sensitivity before as well as after the 12-week intervention (pretrained: *r* = 0.50, p=0.001, and post-trained: *r* = 0.43, p=0.028).

To investigate the potential effect of serum CD300LG on SkM and ScWAT, we performed an overrepresentation analysis on the top 500 mRNAs that were positively correlated (p<0.05) with serum CD300LG levels in each tissue (*Figure 5E–H*). Pathway analyses revealed that the change in serum CD300LG concentrations correlated with changes in expression of genes involved in oxidative phosphorylation, G2M check point and hypoxia both in ScWAT and SkM (*Figure 5E and F*). In ScWAT, serum CD300LG levels also showed the strongest enrichment with angiogenesis pathways (*Figure 5E*). In ScWAT, the change in ScWAT *CD300LG* mRNA levels correlated positively with the change in ScWAT mRNA of genes related to angiogenesis/vasculature development (*Figure 5G*). Similar correlations between *CD300LG* mRNA and angiogenesis genes were observed in SkM as well (*Figure 5H*). For

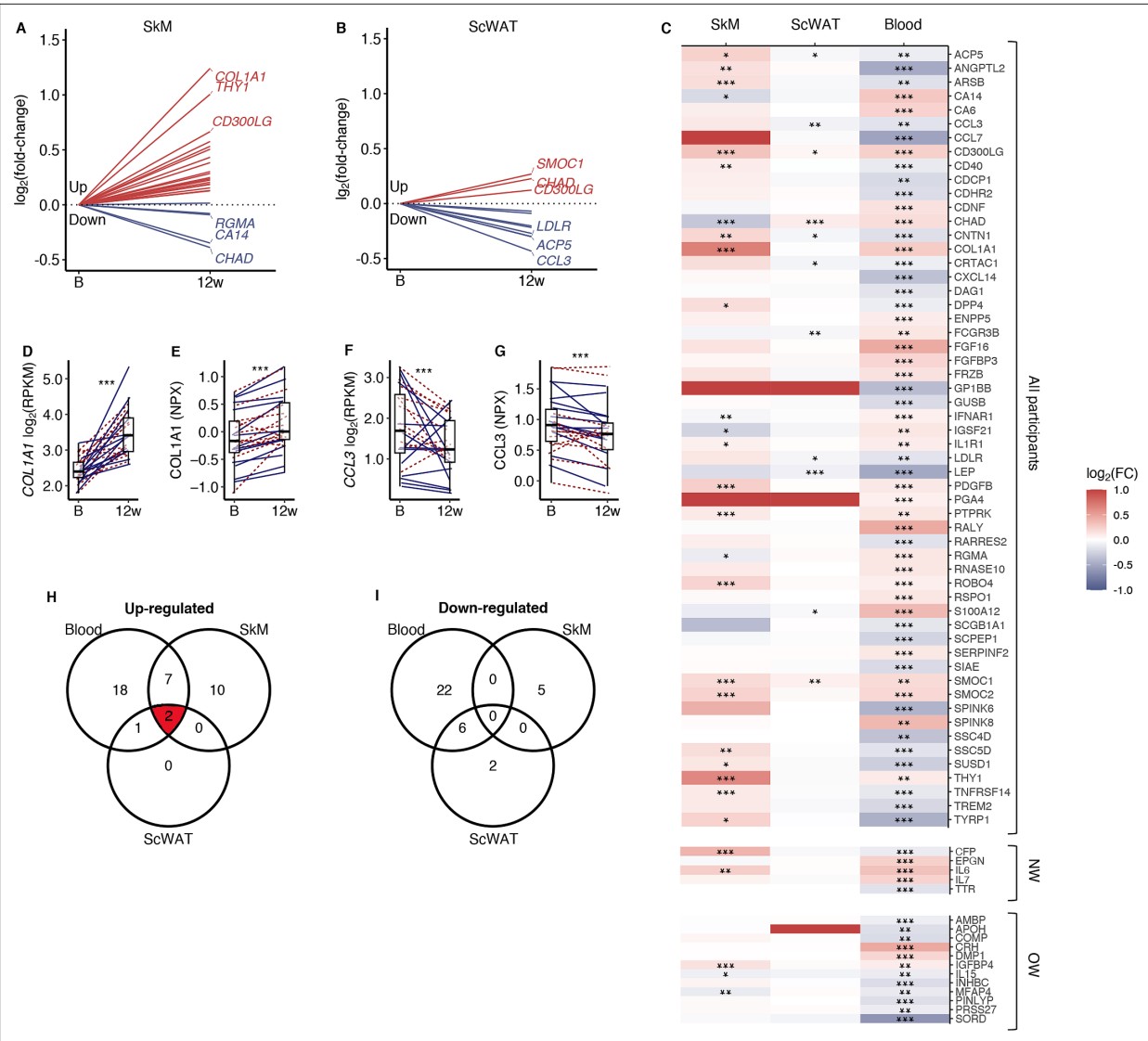

**Figure 4.** Comparison of secretory protein responses to prolonged exercise in blood with corresponding mRNA levels in skeletal muscle and adipose tissue. (**A**) mRNA levels in skeletal muscle and (**B**) adipose tissue for proteins that responded significantly to prolonged exercise. (**C**) A heatmap of log$_2$ (fold-changes) in blood, skeletal muscle, and adipose tissue. (**D**) The most responding mRNA in skeletal muscle, and (**E**) the response in the blood protein. (**F**) The most responding mRNA in adipose tissue, and (**G**) the response in the blood. (**H, I**) Venn diagrams of significant changes in blood, skeletal muscle, and adipose tissue. FC = fold-change; SkM = skeletal muscle; ScWAT = subcutaneous adipose tissue; NPX = normalized protein expression; RPKM = reads per kilobase per million mapped read. *p<0.05, **p<0.01, and ***p<0.001.

example, 60% of the mRNAs in the angiogenesis pathway correlated with *CD300LG* (**Figure 5H**). However, serum CD300LG levels were also correlated positively with pathways related to fatty acid metabolism in both ScWAT (**Figure 5E**) and SkM (**Figure 5H**).

To explore tissue-specific expression of CD300LG, we utilized data from a publicly available human tissue panel (**Uhlén et al., 2015**). CD300LG is highly expressed in adipose tissue compared to other tissues (**Figure 5—figure supplement 1A**), supporting our observation that ScWAT expression was substantially higher than in SkM (**Figure 5B and C**). To further investigate which cells in ScWAT that express CD300LG, we utilized data from a single-cell mRNA sequencing atlas of human ScWAT (https://singlecell.broadinstitute.org/single_cell) generated by **Emont et al., 2022**. *CD300LG* mRNA in ScWAT was primarily expressed in venular endothelial cells, but not adipocytes or other cell types present in ScWAT (**Figure 5—figure supplement 1B–E**).

We next explored whether CD300LG mediates tissue–tissue cross-talk using data from the GD-CAT (Genetically Derived Correlations Across Tissues) database (**Zhou et al., 2024**; **Battle et al., 2017**),

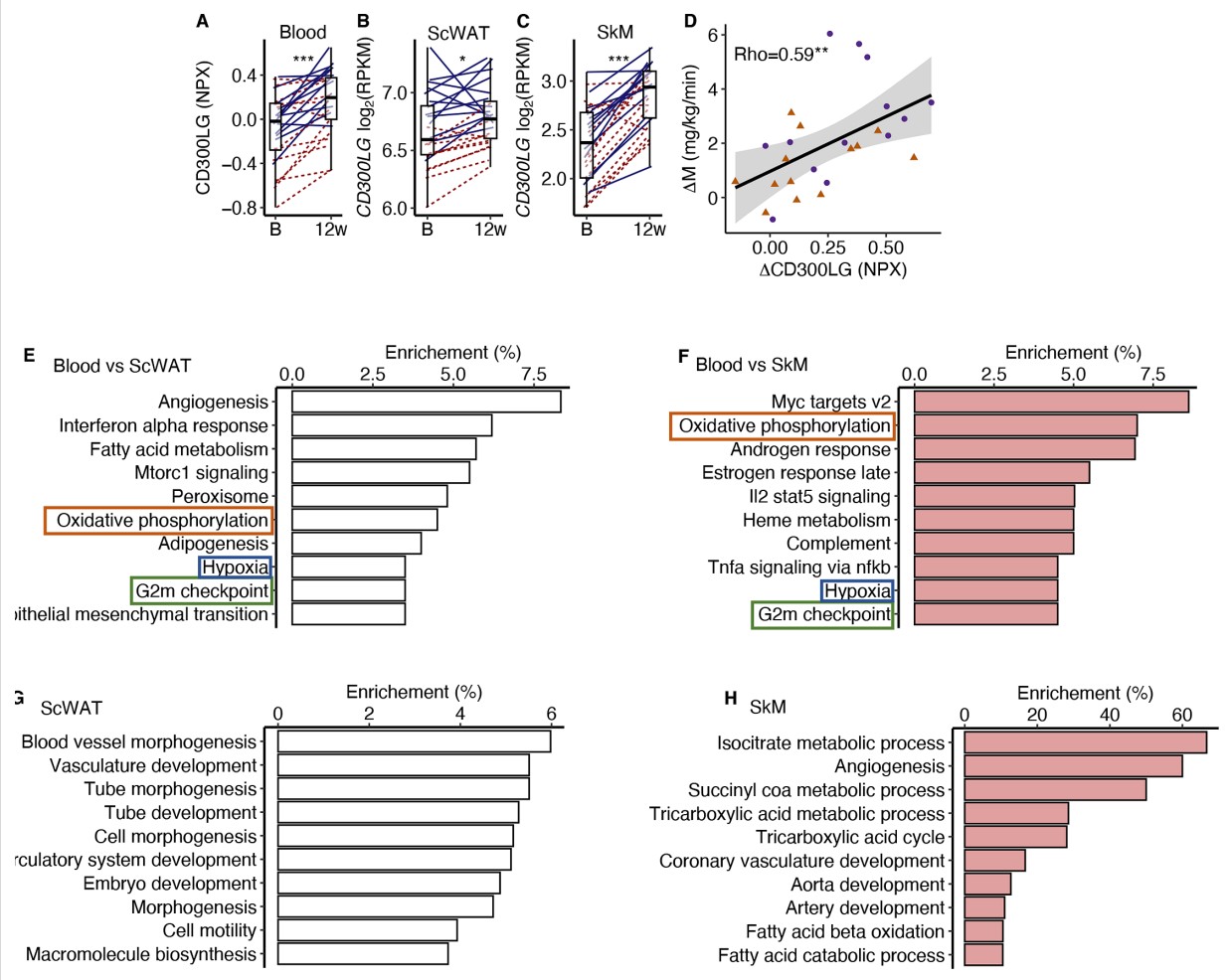

**Figure 5.** CD300LG. (**A**) The response from baseline to week 12 in serum CD300LG and CD300LG mRNA in (**B**) subcutaneous adipose tissue (ScWAT) and (**C**) skeletal muscle (SkM). (**D**) Correlation between the change from before to after prolonged exercise in serum CD300LG and insulin sensitivity. (**E–H**) Pathway enrichment analyses were performed on the top 500 most correlated (and p<0.05) genes in (**E**) ScWAT or (**F**) SkM to the change in serum CD300LG levels, or to the change in CD300LG mRNA levels in (**G**) ScWAT or (**H**) SkM. Only the top 10 pathways with Q < 0.05 are presented. *p<0.05, **p<0.01, and ***p<0.001.

The online version of this article includes the following figure supplement(s) for figure 5:

**Figure supplement 1.** Tissue- and cell-specific expression of CD300LG.

**Figure supplement 2.** *CD300LG* mRNA correlations in men.

**Figure supplement 3.** *CD300LG* mRNA correlations in women.

**Figure supplement 4.** Manhattan plot for serum CD300LG protein levels GWAS.

**Figure supplement 5.** Quantile–quantile plot for CD300LG protein GWAS.

**Figure supplement 6.** Functional validations.

which is a tool for analyzing human gene expression correlations in and across multiple tissues. In men, ScWAT *CD300LG* correlated strongly with ScWAT, SkM, and aortic gene expression (***Figure 5—figure supplement 2***). Consistent with our observations in the MyoGlu exercise intervention study, the top network of gene expression in ScWAT related to ScWAT *CD300LG* mRNA was angiogenesis (***Figure 5—figure supplement 2A***). Like ScWAT, SkM *CD300LG* also correlated strongly with ScWAT, SkM, and aortic gene expression (***Figure 5—figure supplement 2B***). The proteasome complex was the top network of gene expression related to SkM *CD300LG* mRNA (***Figure 5—figure supplement 2B***). In contrast, running the same analyses in women did not reveal associations between *CD300LG* and angiogenesis (***Figure 5—figure supplement 3A and B***).

**Table 1.** Multiple regression analyses between serum CD300LG, and measures of physical activity and glucometabolic traits in the UK Biobank.

| | Women | | | Men | | | Interaction | | | Description | No. of women | No. of men |
|---|---|---|---|---|---|---|---|---|---|---|---|---|
| | Beta-estimate | SE | p | Beta-estimate | SE | p | Beta-estimate | SE | p | | | |
| *NPX ~ physical activity* | | | | | | | | | | | | |
| MET per week all activity | 1.0E-06 | 1.2E-06 | 0.384 | 4.5E-06 | 9.9E-07 | <0.001 | 4.3E-06 | 1.5E-06 | 0.005 | MET minutes per week | 22,527 | 20,726 |
| MET minutes walking | −6.0E-07 | 2.6E-06 | 0.818 | −1.8E-06 | 2.6E-06 | 0.478 | 1.7E-07 | 3.7E-06 | 0.962 | MET minutes per week | 22,527 | 20,726 |
| MET minutes moderate activity | −2.5E-06 | 2.4E-06 | 0.294 | 1.9E-07 | 2.3E-06 | 0.932 | 1.9E-06 | 3.3E-06 | 0.554 | MET minutes per week | 22,527 | 20,726 |
| MET minutes vigorous activity | 1.0E-05 | 2.8E-06 | <0.001 | 2.2E-05 | 2.1E-06 | <2e-16 | 1.5E-05 | 3.5E-06 | <0.001 | MET minutes per week | 22,527 | 20,726 |
| Sedentary overall average | 0.077 | 0.053 | 0.147 | 0.042 | 0.054 | 0.441 | −0.100 | 0.075 | 0.185 | Proportion sedentary activity. | 7430 | 5825 |
| Light overall average | −0.119 | 0.062 | 0.053 | −0.420 | 0.071 | <0.001 | −0.232 | 0.094 | 0.013 | Proportion light activity. | 7430 | 5825 |
| Moderate/vigorous overall average | 0.633 | 0.171 | <0.001 | 1.357 | 0.194 | <0.001 | 1.043 | 0.254 | <0.001 | Proportion moderate/vigorous activity. | 7430 | 5825 |
| IPAQ activity group | 9.6E-03 | 3.8E-03 | 0.012 | 3.1E-02 | 3.8E-03 | <0.001 | 2.6E-02 | 5.4E-03 | <0.001 | IPAQ category | 22,527 | 20,726 |
| Summed days activity | 1.9E-03 | 5.9E-04 | <0.001 | 3.8E-03 | 5.7E-04 | <0.001 | 2.7E-03 | 8.1E-04 | <0.001 | Days performing walking, moderate and vigorous activity | 23,199 | 21,138 |
| Summed minutes activity | −1.8E-05 | 3.0E-05 | 0.550 | 6.4E-05 | 2.7E-05 | 0.017 | 9.9E-05 | 4.0E-05 | 0.013 | Mins performing walking, moderate and vigorous activity | 22,527 | 20,726 |
| Moderate/vigorous recommendation* | 7.5E-03 | 5.6E-03 | 0.186 | 4.8E-02 | 5.8E-03 | <2e-16 | 4.6E-02 | 8.0E-03 | <0.001 | Yes/no | 22,527 | 20,726 |
| Moderate/vigorous walking recommendation* | −1.3E-05 | 7.2E-03 | 0.999 | 4.0E-02 | 7.4E-03 | <0.001 | 4.6E-02 | 1.0E-02 | <0.001 | Yes/no | 22,521 | 20,723 |
| *Trait ~ NPX* | | | | | | | | | | | 28,108 | 23,841 |
| Body fat percentage impedance | −0.137 | 0.051 | 0.007 | −0.552 | 0.052 | <0.001 | −0.375 | 0.073 | <0.001 | Body fat percentage | 28,099 | 23,802 |
| Whole body fat mass impedance | 0.345 | 0.049 | <0.001 | 0.023 | 0.051 | 0.656 | −0.209 | 0.071 | 0.003 | Fat mass (kg) | 28,108 | 23,866 |
| Whole body fat free mass impedance | 0.450 | 0.050 | <0.001 | 1.045 | 0,090 | <0.001 | 0.622 | 0,101 | <0.001 | Fat free mass (kg) | 28,107 | 23,870 |
| Body mass index | - | - | - | - | - | - | - | - | - | kg/m$^2$ | 24,810 | 21,509 |
| Glucose | −0.066 | 0.016 | <0.001 | −0.041 | 0.022 | 0.062 | 0.033 | 0.026 | 0.202 | mmol/L | 27,271 | 23,294 |
| HbA1c | −0.898 | 0.078 | <0.001 | −0.911 | 0.107 | <0.001 | −0.010 | 0.130 | 0.936 | mmol/mol | 27,323 | 23,292 |
| Triglycerides | −0.399 | 0.011 | <0.001 | −0.413 | 0.017 | <0.001 | −0.058 | 0.020 | 0.004 | mmol/L | 28,387 | 24,332 |
| Type 2 diabetes | −0.012 | 0.002 | <0.001 | −0.018 | 0.004 | <0.001 | 0.000 | 0.004 | 0.982 | Yes/no | 24,802 | 21,483 |
| TyG | −2.189 | 0.074 | <0.001 | −2.228 | 0.120 | <0.001 | −0.203 | 0.136 | 0.138 | mmol/L × mmol/L | 22,527 | 23,841 |

Model 1 (NPX ~ physical activity) was a linear regression model of NPX values as a function of a measure of physical activity.

Model 2 (Trait ~ NPX) indicates the measures of body composition and glucometabolic traits were the outcomes and NPX values were set as the exposure.

Models 1 and 2 were adjusted for age, batch, study centre, storage time, and BMI.

*Indicates whether a person met the 2017 UK Physical activity guidelines of 150 min of moderate activity per week or 75 min of vigorous activity.

MET = metabolic equivalent of task. NPX = normalized protein expression. SE = standard error. IPAQ = International Physical Activity Questionnaire. TyG = triglyceride glucose index on insulin resistance.

We then evaluated serum CD300LG levels in up to 47,747 samples in the UK Biobank (see 'Methods'). Descriptive statistics of the UK Biobank cohort are presented in *Supplementary file 1E*. Serum CD300LG levels were positively associated with several measures of physical activity (all metabolic equivalent tasks, results from the international physical activity questionnaire and meeting the recommended amount of weekly physical activity or not; *Table 1*). Interestingly, serum CD300LG

levels were most strongly related to vigorous activity (*Table 1*). Furthermore, the associations between serum CD300LG and physical activity were significantly stronger in men than in women (*Table 1*). Serum CD300LG levels were also positively associated with fat mass and fat free mass, and negatively associated with glucometabolic traits, including serum glucose levels, Hb1Ac, and the risk of having type 2 diabetes (*Table 1*). These associations were independent of BMI.

## GWAS of serum CD300LG levels

GWAS analyses of CD300LG levels detected 43 independent genome-wide significant genetic associations across the genome (*Figure 5—figure supplement 4*). The genomic inflation factor ($\lambda$ = 1.0966) and LD score intercept (1.039) were consistent with our GWAS being well controlled for population stratification and other possible biases (*Figure 5—figure supplement 5*). The most significant SNPs lay along chromosome 17, with these SNPs mapping to the genomic region encoding the *CD300LG* gene (*Figure 5—figure supplement 4*). Follow-up analyses revealed three significant, independent *cis*-pQTLs associated with the protein CD300LG (*Supplementary file 1F*) and a number of *trans*-pQTLs (*Supplementary file 1G*).

## MR analysis

The independent genome-wide significant SNPs from the CD300LG GWAS were used for two-sample MR (see 'Methods' for details), where 39 SNPs were also available in the outcome GWAS (*Chen et al., 2021*). We first performed IVW MR analysis to test the causal relationship between CD300LG and fasting glucose, 2 hr post oral glucose tolerance test (OGTT) glucose levels and HbA1c using only *cis*-SNPs (*Supplementary file 1H*) and all SNPs (*Supplementary file 1I*). The *cis* IVW MR analysis showed some evidence for a negative causal effect of CD300LG on fasting insulin (p=0.01), but due to only three SNPs in these analyses, we could not perform additional sensitivity analyses (except for tests of heterogeneity in estimates of the causal effect across SNPs) and could not determine whether the absence of strong evidence for a causal effect on the glycemic parameters was genuine or whether our analyses just lacked power. Although some of the analyses involving all the genome-wide significant SNPs indicated a potential causal link between increased serum CD300LG concentration and these outcomes (*Supplementary file 1I*), the analysis showed significant heterogeneity. We did not detect strong evidence of directional pleiotropy (significant MR Egger intercept, *Supplementary file 1I*). The heterogeneity in the analysis is possibly due to the fact that many of the SNPs found in the GWAS of CD300LG are associated with related phenotypes that could exert pleiotropic effects on diabetes related outcomes, and so the results should be interpreted with care (*Supplementary file 1J*). Due to the heterogeneity in our results, we therefore performed MR PRESSO to account for outliers. The MR PRESSO analysis showed a significant negative effect of CD300LG on all outcomes (*Table 2*).

**Table 2.** Mendelian randomization (MR) of serum CD300LG levels and glucose outcomes using MR PRESSO.

| Outcome | MR analysis | Number of outliers | Effect | SD | p-value |
|---|---|---|---|---|---|
| 2 hr post OGTT glucose (mmol/L) | Raw | | –0.3722 | 0.0998 | $6.2 \times 10^{-4}$ |
| 2 hr post OGTT glucose (mmol/L) | Outlier-corrected | 2 | –0.3049 | 0.0855 | $1.04 \times 10^{-2}$ |
| Fasting glucose (mmol/L) | Raw | | –0.0307 | 0.0358 | 0.3963 |
| Fasting glucose (mmol/L) | Outlier-corrected | 2 | –0.0556 | 0.0133 | $1.73 \times 10^{-4}$ |
| Fasting insulin (pmol/L) | Raw | | –0.0870 | 0.0558 | 0.1271 |
| Fasting insulin (pmol/L) | Outlier-corrected | 10 | –0.0534 | 0.0252 | 0.0432 |
| HbA1c (%) | Raw | | –0.0485 | 0.0155 | $3.28 \times 10^{-3}$ |
| HbA1c (%) | Outlier-corrected | 3 | –0.0560 | 0.0155 | $1.04 \times 10^{-4}$ |

For detailed results, see **Supplementary file 1H and I**. Fasting glucose adjusted for body mass index (BMI) n = 200,622, 2 hr post oral glucose tolerance test (OGTT) glucose adjusted for BMI n = 63,396, fasting insulin adjusted for BMI n = 15,1013, and HbA1c n = 146,806.

## Mouse models

To functionally validate association of CD300LG with metabolic homeostasis, we leveraged phenotypic data for exercising mice and for *Cd300lg* deficient (*Cd300lg*[-/-]) mice that both were publicly available through the MoTrPAC (*Sanford et al., 2020*) study and the international mouse phenotyping consortium (PhenoMouse) (*Dickinson et al., 2016*).

There is a 51% homology between human *CD300LG* and mouse *Cd300lg* (*Takatsu et al., 2006*), and also in mice, Cd300lg is predominantly expressed in adipose tissue endothelial cells (*Emont et al., 2022*).

In MoTrPAC (*Sanford et al., 2020*), Cd300lg levels in scWAT from n = 12–15 male and female mice were increased after exercise for 8 weeks (~30% in both female [p=0.03] and male [p=0.01] mice) (*Figure 5—figure supplement 6A–C*). Based on data from n = 3050 mice from PhenoMouse male, but not female, mutants for the *Cd300lg*[tm1a(KOMP)Wtsi] allele displayed impaired glucose tolerance (*Figure 5—figure supplement 6D*), but no change in fasting glucose and insulin (*Figure 5—figure supplement 6E and F*). Mutant male, but not female, mice also displayed increased lean mass (*Figure 5—figure supplement 6G*) and less fat mass (*Figure 5—figure supplement 6H*). Detailed PhenoMouse results are presented in *Supplementary file 1K*.

## Discussion

In the present study, we characterized the effects of strength and endurance exercise on the serum proteome of sedentary normal weight and overweight men. We identified significant changes in 283 serum proteins related to many signaling pathways after the 12-week intervention. Some of these proteins were related to the mitochondria, muscle differentiation, and exercise capacity. Among known secretory proteins, 19.7 and 12.1% displayed corresponding mRNA changes in SkM and ScWAT, respectively. Although some proteins may be myokines, others may be adipokines or other types of exerkines. A multi-tissue responding protein was CD300LG, which also correlated positively to insulin sensitivity. CD300LG was particularly interesting because we could replicate the finding in an external cohort, find evidence of a causal link to glucose homeostasis, and perform functionally validation in mice models. Furthermore, the association between CD300LG, physical activity, and glycemic traits might display sex dimorphic relationships.

One of the protein signatures observed in response to exercise was based on strong associations with markers of liver fat content in overweight men. This was related to SLC22A1, which regulates the hepatic glucose fatty acid cycle affecting gluconeogenesis and lipid metabolism (*Liang et al., 2018*), and may influence liver fat accumulation (*Chen et al., 2014*). This signature also shared many common proteins with a known serum NAFLD proteomic signature (*Govaere et al., 2023*). However, we did not detect overlaps between proteins in this signature and specific gene expression patterns of liver cells (e.g., hepatocytes, immune cells) (*Aizarani et al., 2019*). This observation suggests that the proteins detected do not relate to liver protein synthesis per se, but may accumulate in serum due to being released in the blood stream as a result of impaired liver protein catabolism or cell damage as a consequence of overweight/obesity. Notably, this protein signature in overweight men normalized after 12 weeks of exercise and resembled the signature observed in normal weight men. These data suggest prolonged exercise leads to improvements of liver function in overweight men.

Several proteins responding to prolonged exercise had a known signal sequence. These secretory proteins are of particular interest because they could mediate inter-tissue adaptations to exercise. For example, COL1A1 was substantially increased in serum and its corresponding mRNA level was increased in SkM. However, COL1A1 is a collagen peptide that is related to muscle damage, turnover, and extracellular matrix remodeling in response to exercise (*Jacob et al., 2022*) and may mostly reflect muscle restructuring and not represent signaling effects to distant tissues. The large overlap between serum proteins and SkM mRNA most likely suggests a similar phenomenon, where tissue restructuring following exercise is reflected in blood. However, there are probably also several myokines with distant signaling effects among the identified proteins. CCL3 was reduced in serum in parallel with a reduction in its mRNA level in ScWAT. CCL3 is a monocyte chemoattractant protein that may be related to immune cell infiltration in adipose tissue (*Barry et al., 2017*). Hence, this may reflect a positive effect of prolonged exercise on adipose tissue inflammation, which is in line with our

previous results showing normalization of adipose tissue inflammation following prolonged exercise (*Lee et al., 2019*).

A particularly interesting protein was CD300LG, which responded to prolonged exercise in serum, and, judged by its mRNA levels, in SkM and ScWAT. Serum CD300LG levels were lower in overweight compared to normal weight men. Furthermore, the exercise-induced response in CD300LG correlated positively to improvements in insulin sensitivity, and there was also a significant correlation between serum CD300LG and insulin sensitivity both before and after the intervention. We therefore analyzed CD300LG in an external data set, the UK Biobank, and again we observed positive associations between especially vigorous exercise and serum CD300LG. Moreover, serum CD300LG levels were negatively associated with glucose levels and type 2 diabetes in the UK Biobank, and these associations might be causal based on MR analysis. These findings were functionally corroborated by the alterations in glucose tolerance and parameters related to insulin sensitivity observed in *Cd300lg*$^{-/-}$ mice. Thus, CD300LG may represent an exerkine with a causal link to glucose homeostasis. However, whether CD300LG can mediate tissue-tissue crosstalk is unknown. CD300LG is a cell surface protein with a transmembrane domain, but is also a predicted secretory protein (*Meinken et al., 2015*). Whether the protein is released from the cell surface in a regulated manner to mediate cross-tissue signaling needs further investigation. Furthermore, the exact link between CD300LG and glucose metabolism is not clear, but possibly related to the fact that CD300LG is expressed in endothelial cells (*Umemoto et al., 2013*), linked to blood pressure (*Støy et al., 2014*), lymphocyte binding (*Umemoto et al., 2006*), blood triacylglycerol levels (*Surakka et al., 2015*; *Støy et al., 2015*), and molecular traffic across the capillary endothelium (*Takatsu et al., 2006*). Both MyoGlu and GD-CAT data also suggested that CD300LG may be related to angiogenesis in ScWAT (*Van Pelt et al., 2017*) and SkM (*Van Pelt et al., 2017*; *Ross et al., 2023*), at least in men. Hence, we speculate that the link between CD300LG and glucose metabolism is related to improved tissue capillarization/vascular function following prolonged exercise. Furthermore, since vigorous exercise leads to angiogenesis in ScWAT and SkM (*Van Pelt et al., 2017*), serum CD300LG may be a maker of exercise intensity.

## Strengths and limitations

Although MyoGlu included only 26 sedentary men, they were extensively phenotyped with euglycemic hyperinsulinemic clamp, fitness tests, whole body imaging (MRI/MRS), and mRNA sequencing of ScWAT and SkM. We also supplied our study with data from 47,747 persons in the UK Biobank to enhance the validity and generalization of the results. Furthermore, to assess sex differences we stratified analyses for men and women in the UK Biobank, in external data bases (GD-CAT; *Zhou et al., 2024*) and analyzed data from both male and female mice. Since correlations with the clamp data only imply a role for a protein with regard to glucose homeostasis, so we also tested associations with related glucometabolic traits in the UK Biobank and tested these associations for causality using MR. We also included data from exercised mice and mutant mice to further strengthen the results. Our serum proteome study assessed 3072 proteins, and therefore we do not cover the complete human proteome. However, the Olink platform is based on dual recognition of correctly matched DNA-labeled antibodies and DNA sequence-specific protein-to-DNA conversion to generate a signal. This is a highly scalable method with an exceptional specificity (https://olink.com/technology/what-is-pea). Previous exercise-proteomic studies has looked at ~600 proteins in overweight men after endurance exercise (*Diaz-Canestro et al., 2023*), and three papers were published from the HERITAGE study analyzing ~5000 proteins in response to endurance exercise (*Robbins et al., 2021*; *Robbins et al., 2023*; *Mi et al., 2023*). However, our study is the first and largest exercise study using PEA in both overweight and normal weight men, and also including strength exercise.

However, CD300LG's role related to angiogenesis is only suggested through association analyses in our data, necessitating follow-up studies to confirm any causal role of CD300LG in angiogenesis. Although the open-source cd300lgtm1a(KOMP)Wtsi mice provided interesting indications, a future study should directly phenotype mice with alterations in the CD300LG gene and measure the effects on circulating CD300LG levels and potential regulatory mechanisms related to angiogenesis and glucose tolerance. Furthermore, it is also unknown if circulating CD300LG is full-length or a cleaved fragment, and the mechanisms for CD300LG secretion should be further studied in vitro. Finally, future experiments should also identify the epitope for O-link binding, and confirm its specificity using targeted mass spectrometry or antibody-based validations.

## Conclusion

Our study provided a detailed analysis of serum proteins responding to 3 months of strength and endurance exercise in both normal weight and overweight men. Our results identified a novel NAFLD-related serum protein signature in overweight men that was normalized after prolonged exercise. We also identified hundreds of tissue-specific and multi-tissue serum markers of, for example,, mitochondrial function, muscle differentiation, exercise capacity, and insulin sensitivity. Our results were enriched for secretory proteins (exerkines), such as CD300LG, which may be a marker of exercise intensity especially in men, and may also have causal roles in improved glucose homeostasis after physical activity.

## Additional information

### Competing interests

Marcus M Seldin: Reviewing editor, *eLife*. Christian A Drevon: Stock owner at VITAS AS. The other authors declare that no competing interests exist.

### Funding

| Funder | Grant reference number | Author |
|---|---|---|
| South-Eastern Norway Regional Health Authority | | Kåre Inge Birkeland |
| Simon Fougners Fund | | Kåre Inge Birkeland |
| Diabetesforbundet | | Sindre Lee-Ødegård |
| Norwegian Medical Association | Johan Selmer Kvanes' legat til forskning og bekjempelse av sukkersyke | Sindre Lee-Ødegård |
| UK Biobank | 53641 | David M Evans |
| Australian National Health and Medical Research Council | APP2017942 | David M Evans |
| Australian Research Council | DE220101226 | Gunn-Helen Moen |
| Research Council of Norway | 325640 | Gunn-Helen Moen |
| Research Council of Norway | 287198 | Gunn-Helen Moen |
| University of Oslo | Medical Student Research Program | Jonas Krag Viken |
| Novo Nordisk Fonden | NNF23OC0082123 | Sindre Lee-Ødegård |

The funders had no role in study design, data collection and interpretation, or the decision to submit the work for publication.

### Author contributions

Sindre Lee-Ødegård, Conceptualization, Resources, Data curation, Formal analysis, Funding acquisition, Validation, Investigation, Visualization, Methodology, Writing - original draft, Project administration, Writing – review and editing; Marit Hjorth, Data curation, Visualization, Methodology, Writing – review and editing; Thomas Olsen, Jonas Krag Viken, Writing – review and editing; Gunn-Helen Moen, Emily Daubney, Resources, Formal analysis, Visualization, Methodology, Writing – review and editing; David M Evans, Resources, Supervision, Visualization, Methodology, Writing – review and editing; Andrea L Hevener, Resources, Software, Supervision, Writing – review and editing; Aldons J Lusis, Resources, Supervision, Writing – review and editing; Mingqi Zhou, Resources, Software, Methodology, Writing – review and editing; Marcus M Seldin, Resources, Software, Writing – review and editing; Hooman Allayee, Resources, Data curation, Software, Supervision, Validation, Investigation,

Visualization, Methodology, Writing – review and editing; James Hilser, Validation, Visualization, Methodology, Writing – review and editing; Hanne Gulseth, Data curation, Methodology, Writing – review and editing; Frode Norheim, Supervision, Writing – review and editing; Christian A Drevon, Conceptualization, Supervision, Funding acquisition, Methodology, Writing – review and editing; Kåre Inge Birkeland, Conceptualization, Supervision, Funding acquisition, Project administration, Writing – review and editing

### Author ORCIDs

Sindre Lee-Ødegård ⬤ https://orcid.org/0000-0002-0670-7555
Thomas Olsen ⬤ https://orcid.org/0000-0003-1805-5221
David M Evans ⬤ https://orcid.org/0000-0003-0663-4621
Aldons J Lusis ⬤ https://orcid.org/0000-0001-9013-0228
Mingqi Zhou ⬤ https://orcid.org/0009-0007-7643-7873
Marcus M Seldin ⬤ https://orcid.org/0000-0001-8026-4759
Hooman Allayee ⬤ https://orcid.org/0000-0002-2384-5239
Christian A Drevon ⬤ https://orcid.org/0000-0002-7216-2784

### Ethics

Clinical trial registration clinicaltrials.gov: NCT01803568.

The MyoGlu study was conducted as a controlled clinical trial (clinicaltrials.gov: NCT01803568) and was carried out in adherence to the principles of the Declaration of Helsinki. The study received ethical approval from the National Regional Committee for Medical and Health Research Ethics North in Tromsø, Norway, with the reference number 2011/882. All participants provided written informed consent before undergoing any procedures related to the study. The UK Biobank has ethical approval from the North West Multi Centre Research Ethics Committee (MREC), which covers the UK, and all participants provided written informed consent. This particular project from the UK Biobank received ethical approval from the Institutional Human Research Ethics committee, University of Queensland (approval number 2019002705).

Reviewer #1 (Public Review): https://doi.org/10.7554/eLife.96535.3.sa1
Reviewer #2 (Public Review): https://doi.org/10.7554/eLife.96535.3.sa2
Author response https://doi.org/10.7554/eLife.96535.3.sa3

---

# Additional files

### Supplementary files

• MDAR checklist

• Supplementary file 1. Detailed results. (A) Subject characteristics at baseline and changes observed after 12 weeks of exercise. (B) All mixed model results for Olink proteins in response to prolonged exercise. (C) Common and specific upregulated proteins in response to prolonged exercise in all participants, only normal weight men and in only overweight men. (D) Common and specific downregulated proteins in response to prolonged exercise in all participants, only normal weight men and in only overweight men. (E) Descriptive statistics UK Biobank. (F) Serum CD300LG cis-pQTLs. (G) Serum CD300LG trans-pQTLs. (H) Results of cis-only pQTLs MR analysis for serum protein level (log$_2$) on outcomes of interest (2-hr post OGTT glucose [mmol/L], fasting glucose [mmol/L], fasting insulin [mmol/L], and HbA1c [%]). (I) Results of cis- and trans-combined pQTLs MR analysis for outcomes of interest (2-hr post OGTT glucose, fasting glucose, fasting insulin, and HbA1c). (J) SNPs and their prior associations. (K) All available phenotype data on the Cd300lg knock-out mice model.

### Data availability

mRNA sequencing data from MyoGlu can be found at https://exchmdpmg.medsch.ucla.edu/app/ as well as in GSE227419. Secretory proteins are available in the MetazSecKB data base at http://proteomics.ysu.edu/secretomes/animal/. The human serum proteomic NAFLD signature is available in the study of *Govaere et al., 2023*. Expression profiles in human liver cells are available in the Human Liver Cell Atlas *Aizarani et al., 2019*. Data obtained from the UK biobank (Olink explore 1536 and measures of physical activity *Cassidy et al., 2016* can be found at https://biobank.ndph.ox.ac.

uk/ukb/. Glucometabolic outcomes used in MR analyses are available at: http://magicinvestigators. org/ *Chen et al., 2021*. Data from the GD-CAT database *Zhou et al., 2024* is available from: https:// pipeline.biochem.uci.edu/gtex/demo2/. Mice exercise data are available at https://motrpac-data.org/ and knock-out data at https://www.mousephenotype.org/. CD300LG expression values from a human tissue panel were obtained from *Uhlén et al., 2015*. The single nuclei mRNA sequencing data from human adipose tissue was plotted in Seurat v. 4 by downloading processed data from the Single Cell Portal *Emont et al., 2022*. The data can also be explored at: https://singlecell.broadinstitute. org/single_cell/study/SCP1376/a-single-cell-atlas-of-human-and-mouse-white-adipose-tissue). UK Biobank (https://www.ukbiobank.ac.uk/) data are available to researchers upon application to the individual cohorts via their websites. All other data used are publicly available and referenced according in the main text. For additional details and data inquiries, please contact Sindre Lee-Ødegård.

The following dataset was generated:

| Author(s) | Year | Dataset title | Dataset URL | Database and Identifier |
|-----------|------|---------------|-------------|------------------------|
| Lee-Ødegård S | 2024 | Serum proteomic profiling of physical activity reveals CD300LG as a novel exerkine with a potential causal link to glucose homeostasis | http://www.ncbi. nlm.nih.gov/geo/ query/acc.cgi?acc= GSE227419 | NCBI Gene Expression Omnibus, GSE227419 |

The following previously published datasets were used:

| Author(s) | Year | Dataset title | Dataset URL | Database and Identifier |
|-----------|------|---------------|-------------|------------------------|
| Meinken J, Walker G, Cooper CR, Min XJ | 2015 | MetazSecKB is a knowledgebase for human/ animal secretomes as well as human/animal proteins located in other subcellular locations | http://proteomics. ysu.edu/secretomes/ animal/index.php | Database Commons, MetazSecKB |

*Continued on next page*

*Continued*

| Author(s) | Year | Dataset title | Dataset URL | Database and Identifier |
|---|---|---|---|---|
| Sun BB, Chiou J, Traylor M, Benner C, Hsu YH, Richardson TG, Surendran P, Mahajan A, Robins C, Vasquez-Grinnell SG, Hou L, Kvikstad EM, Burren OS, Davitte J, Ferber KL, Gillies CE, Hu S, Lin T, Mikkilineni R, Pendergrass RK, Pickering C, Prins B, Baird D, Chen CY, Ward LD, Deaton AM, Welsh S, Willis CM, Lehner N, Arnold M, Wörheide MA, Suhre K, Kastenmüller G, Sethi A, Cule M, Raj A, Alnylam Human Genetics, AstraZeneca Genomics Initiativ, Biogen Biobank Team, Squibb BM, Genentech Human Genetics, GlaxoSmithKline Genomic Sciences, Pfizer Integrative Biology, Population Analytics of Janssen Data Sciences, Regeneron Genetics Center, Burkitt-Gray L, Melamud E, Black ME, Fauman EB, Howson JMM, Kang HM, McCarthy MI, Nioi P, Petrovski S, Scott RA, Smith EN, Szalma S, Waterworth DM, Mitnaul LJ, Szustakowski JD, Gibson BW, Miller MR, Whelan CD | 2023 | Plasma proteomic associations with genetics and health in the UK Biobank | https://biobank.ndph. ox.ac.uk/showcase/ label.cgi?id=1839 | UK Biobank, 1839 |
| Emont MP, Jacobs C, Essene AL, Pant D, Tenen D, Colleluori G, Di Vincenzo A, Jørgensen AM, Dashti H, Stefek A, McGonagle E, Strobel S, Laber S, Agrawal S, Westcott GP, Kar A, Veregge ML, Gulko A, Srinivasan H, Kramer Z, De Filippis E, Merkel E, Ducie J, Boyd GC, Gourash W, Courcoulas A, Lin SJ, Lee BT, Morris D, Tobias A, Khera AV, Claussnitzer M, Pers TH, Giordano A, Ashenberg O, Regev A, Tsai LT, Rosen ED | 2022 | A single cell atlas of human and mouse white adipose tissue | https://singlecell. broadinstitute.org/ single_cell/study/ SCP1376/a-single- cell-atlas-of-human- and-mouse-white- adipose-tissue | Single Cell Portal, SCP1376 |

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
