## [Editor Report · eLife assessment]

This **useful** article describes a proteomic analysis of plasma from subjects before and after an exercise regime consisting of endurance and resistance exercise. The work identifies a putative new exerkine, CD300LG, and finds associations of this protein with aspects of insulin sensitivity and angiogenesis. The characterization remains **incomplete** at present. Because CD300LG may have a transmembrane domain, one possibility is that exercise causes the release of extracellular vesicles containing this protein. As this study reports associations, additional studies will be needed to establish causality. The article will hopefully prompt further studies to more fully elucidate the underlying biology.

---

## [Referee Report · Reviewer #1 (Public Review)]

Summary:

In this paper, proteomics analysis of the plasma of human subjects that underwent an exercise training regime consisting of a combination of endurance and resistance exercise led to the identification of several proteins that were responsive to exercise training. Confirming previous studies, many exercise-responsive secreted proteins were found to be involved in the extra-cellular matrix. The protein CD300LG was singled out as a potential novel exercise biomarker and the subject of numerous follow-up analyses. The levels of CD300LG were correlated with insulin sensitivity. The analysis of various open-source datasets led to the tentative suggestion that CD300LG might be connected with angiogenesis, liver fat, and insulin sensitivity. CD300LG was found to be most highly expressed in subcutaneous adipose tissue and specifically in venular endothelial cells. In a subset of subjects from the UK Biobank, serum CD300LG levels were positively associated with several measures of physical activity - particularly vigorous activity. In addition, serum CD300LG levels were negatively associated with glucose levels and type 2 diabetes. Genetic studies hinted at these associations possibly being causal. Mice carrying alterations in the CD300LG gene displayed impaired glucose tolerance, but no change in fasting glucose and insulin. Whether the production of CD300LG is changed in the mutant mice is unclear.

Strengths:

The specific proteomics approach conducted to identify novel proteins impacted by exercise training is new. The authors are resourceful in the exploitation of existing datasets to gain additional information on CD300LG.

Weaknesses:

While the analyses of multiple open-source datasets are necessary and useful, they lead to relatively unspecific correlative data that collectively insufficiently advance our knowledge of CD300LG and merely represent the starting point for more detailed investigations. Additional more targeted experiments of CD300LG are necessary to gain a better understanding of the role of CD300LG and the mechanism by which exercise training may influence CD300LG levels. One should also be careful to rely on external data for such delicate experiments as mouse phenotyping. Can the authors vouch for the quality of the data collected?

---

## [Referee Report · Reviewer #2 (Public Review)]

Summary:

This manuscript from Lee-Odegard et al reports proteomic profiling of exercise plasma in humans, leading to the discovery of CD300LG as a secreted exercise-inducible plasma protein. Correlational studies show associations of CD300LG with glycemic traits. Lastly, the authors query available public data from CD300LG-KO mice to establish a causal role for CD300LG as a potential link between exercise and glucose metabolism. However, the strengths of this manuscript were balanced by the moderate to major weaknesses. Therefore in my opinion, while this is an interesting study, the conclusions remain preliminary and are not fully supported by the experiments shown so far.

Strengths:

(1) Data from a well-phenotyped human cohort showing exercise-inducible increases in CD300LG.

(2) Associations between CD300LG and glucose and other cardiometabolic traits in humans, that have not previously been reported.

(3) Correlation to CD300LG mRNA levels in adipose provides additional evidence for exercise-inducible increases in CD300LG.

Weaknesses:

(1) CD300LG is by sequence a single-pass transmembrane protein that is exclusively localized to the plasma membrane. How CD300LG can be secreted remains a mystery. More evidence should be provided to understand the molecular nature of circulating CD300LG. Is it full-length? Is there a cleaved fragment? Where is the epitope where the o-link is binding to CD300LG? Does transfection of CD300LG to cells in vitro result in secreted CD300LG?

(2) There is a growing recognition of specificity issues with both the O-link and somalogic platforms. Therefore it is critical that the authors use antibodies, targeted mass spectrometry, or some other methods to validate that CD300LG really is increased instead of just relying on the O-link data.

(3) It is insufficient simply to query the IMPC phenotyping data for CD300LG; the authors should obtain the animals and reproduce or determine the glucose phenotypes in their own hands. In addition, this would allow the investigators to answer key questions like the phenotype of these animals after a GTT, whether glucose production or glucose uptake is affected, whether insulin secretion in response to glucose is normal, effects of high-fat diet, and other standard mouse metabolic phenotyping assays.

(4) I was unable to find the time point at which plasma was collected at the 12-week time point. Was it immediately after the last bout of exercise (an acute response) or after some time after the training protocol (trained state)?

---

## [Author Response]

The following is the authors’ response to the original reviews.

**Reviewer #1 (Public Review):**
Summary:In this paper, proteomics analysis of the plasma of human subjects that underwent an exercise training regime consisting of a combination of endurance and resistance exercise led to the identification of several proteins that were responsive to exercise training. Confirming previous studies, many exercise-responsive secreted proteins were found to be involved in the extra-cellular matrix. The protein CD300LG was singled out as a potential novel exercise biomarker and the subject of numerous follow-up analyses. The levels of CD300LG were correlated with insulin sensitivity. The analysis of various open-source datasets led to the tentative suggestion that CD300LG might be connected with angiogenesis, liver fat, and insulin sensitivity. CD300LG was found to be most highly expressed in subcutaneous adipose tissue and specifically in venular endothelial cells. In a subset of subjects from the UK Biobank, serum CD300LG levels were positively associated with several measures of physical activity - particularly vigorous activity. In addition, serum CD300LG levels were negatively associated with glucose levels and type 2 diabetes. Genetic studies hinted at these associations possibly being causal. Mice carrying alterations in the CD300LG gene displayed impaired glucose tolerance, but no change in fasting glucose and insulin. Whether the production of CD300LG is changed in the mutant mice is unclear.Strengths:The specific proteomics approach conducted to identify novel proteins impacted by exercise training is new. The authors are resourceful in the exploitation of existing datasets to gain additional information on CD300LG.Weaknesses:While the analyses of multiple open-source datasets are necessary and useful, they lead to relatively unspecific correlative data that collectively insufficiently advance our knowledge of CD300LG and merely represent the starting point for more detailed investigations. Additional more targeted experiments of CD300LG are necessary to gain a better understanding of the role of CD300LG and the mechanism by which exercise training may influence CD300LG levels. One should also be careful to rely on external data for such delicate experiments as mouse phenotyping. Can the authors vouch for the quality of the data collected.

Thank you for the valuable feedback on our manuscript. We recognize concerns about the specificity of correlative data from open-source datasets and the limitations it presents for understanding CD300LG's role. To address this, we have expanded the manuscript with a paragraph in the discussion regarding the need of targeted experiments confirm CD300LG’s functions and relationship with glucose metabolism. We also emphazise caution regarding external data reliance and we acknowledge the need for generating primary data including direct phenotyping of mice with CD300LG gene alterations to better understand its regulatory mechanisms and effects on glucose tolerance. Please see lines 446-456.

**Reviewer #2 (Public Review):**
Summary:This manuscript from Lee-Odegard et al reports proteomic profiling of exercise plasma in humans, leading to the discovery of CD300LG as a secreted exercise-inducible plasma protein. Correlational studies show associations of CD300LG with glycemic traits. Lastly, the authors query available public data from CD300LG-KO mice to establish a causal role for CD300LG as a potential link between exercise and glucose metabolism. However, the strengths of this manuscript were balanced by the moderate to major weaknesses. Therefore in my opinion, while this is an interesting study, the conclusions remain preliminary and are not fully supported by the experiments shown so far.Strengths:(1) Data from a well-phenotyped human cohort showing exercise-inducible increases in CD300LG.(2) Associations between CD300LG and glucose and other cardiometabolic traits in humans, that have not previously been reported.(3) Correlation to CD300LG mRNA levels in adipose provides additional evidence for exercise-inducible increases in CD300LG.Weaknesses:(1) CD300LG is by sequence a single-pass transmembrane protein that is exclusively localized to the plasma membrane. How CD300LG can be secreted remains a mystery. More evidence should be provided to understand the molecular nature of circulating CD300LG. Is it full-length? Is there a cleaved fragment? Where is the epitope where the o-link is binding to CD300LG? Does transfection of CD300LG to cells in vitro result in secreted CD300LG?(2) There is a growing recognition of specificity issues with both the O-link and somalogic platforms. Therefore it is critical that the authors use antibodies, targeted mass spectrometry, or some other methods to validate that CD300LG really is increased instead of just relying on the O-link data.(3) It is insufficient simply to query the IMPC phenotyping data for CD300LG; the authors should obtain the animals and reproduce or determine the glucose phenotypes in their own hands. In addition, this would allow the investigators to answer key questions like the phenotype of these animals after a GTT, whether glucose production or glucose uptake is affected, whether insulin secretion in response to glucose is normal, effects of high-fat diet, and other standard mouse metabolic phenotyping assays.(4) I was unable to find the time point at which plasma was collected at the 12-week time point. Was it immediately after the last bout of exercise (an acute response) or after some time after the training protocol (trained state)?

We acknowledge the importance of understanding the molecular form of CD300LG in circulation. We have expanded the discussion with a paragraph regarding the need of follow-up experiments on whether circulating CD300LG is full-length or a cleaved fragment, to identify the epitope for O-link binding, and assess CD300LG secretion in vitro through transfection experiments. We also discuss the need of targeted mass spectrometry and antibody-based validation of O-link measurements of CD300LG, and the need for more validation experiments on CD300LG-deficient mice. Please see lines 446-456.

The plasma collected post-intervention is in a state that reflects the new baseline trained condition of the subjects, 3 days after the last exercise session during the intervention. We have clarified this in our manuscript. The information is updated in line 491-493.

**Reviewer #1 (Recommendations For The Authors):**
In the present form, the paper raises interest in the potential role of CD300LG in the response to exercise training but unfortunately does not provide clear answers. The authors should focus their efforts on firmly validating the status of CD300LG as an exercise biomarker in humans and carefully examine the function of CD300LG through mechanistic and animal-based studies.The authors are encouraged to acquire CD300LG-deficient mice and perform specific experiments to validate hypotheses forthcoming from the analysis of the open-source datasets. In addition, it needs to be validated that the cd300lgtm1a(KOMP)Wtsi mice are actually deficient in CD300LG. It is not uncommon that Tm1a mice have (almost) normal expression of the targeted gene.

We have now revised the manuscript and added a new section to the discussion regarding the limitations with open-source data, cd300lgtm1a(KOMP)Wtsi mice and the need for more validation experiments on CD300LG-deficient mice. Please see lines 446-456.

The value of the correlative data presented in Figure 5 is rather limited. The same can be argued for the data presented in Supplementary Figure 2. If CD300LG is expressed in endothelial cells, it stands to reason that its expression is correlated with angiogenesis. Hence, this observation does not really carry any additional value.

We agree that correlations cannot imply causality. However, similar patterns were observed in several tissues and across different data sets, which at least suggest a role CD300LG related to angiogesis. We have included a section in the discussion were we clarify that our observations should only be regarded as indications and that follow-up studies are needed to confirm any causal role for CD300LG on angiogenesis/oxidativ capacity. Please see lines 446-456.

Figure 6 may be better accommodated in the supplement.

Figure 6 is now moved to the supplement.

Figure 3A and B are a bit awkward. The description "no overlap" is confusing. Isn't it more accurate to say "no enrichment" or "no over-representation"? There will always be some overlap with certain pathways. However, there may be no enrichment. Furthermore, the use of arrows to indicate No overlap is visually not very appealing. Maybe the numbers can be given a specific color?

We have now removed the arrows and text, and rather stated in the text that there were no enrichements other than for the proteins down-regulated in the overweight group.

The description of the figure legend of figure 5E-H is incomplete.

The description is now completed.